# CT Texture Analysis Challenges: Influence of Acquisition and Reconstruction Parameters: A Comprehensive Review

**DOI:** 10.3390/diagnostics10050258

**Published:** 2020-04-28

**Authors:** Mathilde Espinasse, Stéphanie Pitre-Champagnat, Benoit Charmettant, Francois Bidault, Andreas Volk, Corinne Balleyguier, Nathalie Lassau, Caroline Caramella

**Affiliations:** 1BioMaps, Université Paris-Saclay, CEA, CNRS UMR 9011, Inserm UMR1281, Institut Gustave Roussy, 114 Rue Edouard Vaillant, 94800 Villejuif, France; mathilde.espinasse@gustaveroussy.fr (M.E.); stephanie.champagnat-pitre@gustaveroussy.fr (S.P.-C.); benoit.charmettant@gustaveroussy.fr (B.C.); francois.bidault@gustaveroussy.fr (F.B.); andreas.volk@gustaveroussy.fr (A.V.); corinne.balleyguier@gustaveroussy.fr (C.B.); caroline.caramella@gustaveroussy.fr (C.C.); 2École Normale Supérieure Paris-Saclay, 61 avenue du Président Wilson, 94235 Cachan CEDEX, France; 3Department of Imaging, Gustave Roussy Cancer Campus, 114 rue Edouard Vaillant, 94805 Villejuif CEDEX, France; 4Research Department, Gustave Roussy Cancer Campus, 114 avenue Edouard Vaillant, 94805 Villejuif, France

**Keywords:** radiomics, texture analysis, computed tomography, acquisition parameters

## Abstract

Texture analysis in medical imaging is a promising tool that is designed to improve the characterization of abnormal images from patients, to ultimately serve as a predictive or prognostic biomarker. However, the nature of image acquisition itself implies variability in each pixel/voxel value that could jeopardize the usefulness of texture analysis in the medical field. In this review, a search was performed to identify current published data for computed tomography (CT) texture reproducibility and variability. On the basis of this analysis, the critical steps were identified with a view of using texture analysis as a reliable tool in medical imaging. The need to specify the CT scanners used and the associated parameters in published studies is highlighted. Harmonizing acquisition parameters between studies is a crucial step for future texture analysis.

## 1. Introduction

Computed tomographic (CT) images are routinely used for the diagnosis and follow-up of patients. These images represent a huge amount of numerical data, which are both transferable and storable. 

Recently, the temptation of using these data as potential biomarkers for a large range of clinical questions, especially the prediction of response and prognosis, has modified the way researchers are bethinking imaging. Texture analysis is based on many well described mathematical approaches such as first and second order texture calculation, combined with the application of different mathematical filters, which can lead to up to 1000 texture features for one image. Translating the medical imaging numerical datasets into quantitative figures, and thus authorizing statistical comparisons, is known as radiomics [1]. Numerous publications have already studied texture as a diagnosis or prognosis biomarker, but results are very heterogeneous and confusing [2,3]. Indeed, this new field of research comes with a new language, new concepts and a lack of international standardization.

Although efforts from several international communities, such as the Quantitative Imaging Biomarkers Alliance (QIBA) [4], the Quantitative Imaging Network (QIN) [5] or the Image Biomarker Standardisation Initiative, exist [6], this review is aimed at helping radiologists to better understand the current scientific data specifically based on CT texture variability, and to point out the many challenges texture applied to medical imaging has to face before it could become a reliable biomarker.

## 2. Methods

For the purpose of this work, articles were selected through a PubMed search, using the keywords (“computed tomography” OR “CT”) AND (“texture” OR “radiomics”). A cut-off of 5 February 2018 was used and 1143 articles were identified. 

Studies dealing with the prognosis, diagnosis and responses to treatment of patients, focusing on other imaging techniques than CT or using only first order texture features (histograms) were discarded. Articles retained included patients or phantom studies focusing on the impact of the acquisition and processing parameters on the variability, reproducibility and repeatability of texture features. Finally, 20 articles were included in the study. Table 1 summarises the main characteristics of the selected articles.

The texture features studied in the selected articles included first order indices and higher order features. First order features are parameters extracted from the histogram of the distribution of the values of pixels. Second order features come from the matrices describing the spatial relations between pixels: grey level co-occurrence matrix (GLCM), grey level run length matrix (GLRLM), neighbourhood grey level difference matrix (NGLDM), grey level size zone matrix (GLSZM), grey level zone length matrix (GLZLM), etc. Texture features can also come from fractal or wavelet techniques or Gaussian Markov random fields. 

Articles often dealt with both patients and phantoms, or with more than one parameter; this implies that the same article can be found in numerous parts of this paper. 

## 3. Results

### 3.1. Texture Processing

#### 3.1.1. Software

The choice of software is linked with the method selected. Numerous texture analysis software exist, for example LIFEx [27], IBEX [28], Pyradiomics [29] or MaZda [30]. These software tend to generate a large number of texture features, of which many are common to all software, but not all studies use the same descriptors which makes it difficult to compare the results. Furthermore, it is important to note that sometimes the same name of texture feature can cover different computation methods or different feature names can actually represent the same quantity as described by Buvat et al. [31]. Numerous teams also developed their own in-house software. In order to deal with this issue, developers need to align with current recommendations offered by the Image Biomarker Standardisation Initiative [6] which provides standardized nomenclature and definitions, a standardized image processing workflow and implements guidelines for conducting radiomics studies.

#### 3.1.2. Dimensionality

Features can be calculated in 2, 2.5 or 3 dimensions. These different computation methods give different results, as highlighted by Balagurunathan et al. [8]. They analyse 219 3D texture features and 110 2D with scans from 32 patients from the RIDER database, and they conclude that 3D features better describe the volume but 2D features are more easily interpreted. Fave et al. [12] computed 23 texture features both in 3D and in 2D on the largest cross-sectional slice of patients’ CT scans: 8 varied significantly between 2D and 3D, but 14 were significantly correlated between 2D and 3D with a Spearman correlation coefficient over 0.85. They concluded that the majority of 2D and 3D features translate the same heterogeneity, but that these two computation methods cannot be mixed as the numerical results are different. Despite the fact that the values derived from an analysis of the largest cross-sectional slice seem to be an effective substitute for a whole tumour analysis, they recommend the whole tumour analysis whenever possible to avoid any bias induced by the choice of the slice. 

### 3.2. Texture Repeatability

#### 3.2.1. Intra-Patients Repeatability

Balagurunathan et al. [8] studied the RIDER database which contains repeat CT scans of lung cancer patients performed with the same CT scanner with a 15-minute interval and reported 48 reproducible features (concordance correlation coefficient (CCC) > 0.9) out of 219. Fave et al. [13] extracted texture features from the CBCT scans of 10 patients and excluded 23 features out of 68 as non-reproducible (CCC ≤ 0.9). They also demonstrated with a dynamic-motion thorax phantom, that out of 68 features, 12 are reproducible with a 4 mm movement, and only 3 with a 6 to 8 mm movement.

#### 3.2.2. Phantoms

The majority of repeatability studies are performed on phantoms. Caramella et al. [11] conducted eight consecutive CT scans on the same in-house phantom with the same CT scanner using the same parameters. They extracted 34 features with LIFEx and kept only 8 as reproducible. This emphasizes a lack of experimental reproducibility under the same experimental conditions, which might be of greater concerns in vivo. Berenguer et al. [9] also tested the reproducibility and redundancy of texture features computed with IBEX with test–retest, intra-CT and inter-CT analyses. The CT acquisition parameters remained identical in the test–retest and inter-CT analyses. They found 161 features out of 177 as reproducible, but the redundancy study concluded that the 177 studied features could be summarized by 10 of them. The study acknowledged that multicentre reproducibility is of a great challenge but it can be minimized using rigorous acquisition protocols.

This partial non-reproducibility of texture features may be related to stochastic noise. Al-Kadi et al. [7] studied the impact of various distributional noise on 74 texture features from 7 different computation methods (including different matrices, wavelets and fractal dimension). The enhanced and unenhanced CT scans of 67 patients, taken on 2 different CT scans of lungs showing tumours at different stages were used. The study concluded that they were affected by noise, but differently for each feature. The features with the highest characterization power were the least affected by noise. They showed that adaptive filtering can help reduce subtle noise.

### 3.3. Intrinsic CT Parameters

#### 3.3.1. CT Scanner Brand

Each manufacturer uses its own X-ray tube, detectors, reconstruction and post-processing algorithms to build the image.

Mackin et al. [18] and Fave et al. [13] studied the influence of CT scanners on texture features with the same in-house phantom containing cartridges of different materials (later referred to as the CCR phantom). Mackin et al. compared the interscanner and interpatient variabilities of texture features on 16 CT scanners from four different manufacturers, each with its own standard acquisition protocol. They found that interscanner variability depends on the feature under consideration and the material of the region of interest, but showed that the interscanner are of the same order of magnitude than the interpatient variabilities. Fave et al. scanned the CCR phantom on two cone beam CT scanners, using different acquisition parameters. They got a good reproducibility of features when comparing CT scans acquired from the same manufacturer, whereas using different protocols limited the reproducibility, and comparing the different manufacturers completely withdrew reproducibility. 

Larue et al. [16] studied a modified CCR phantom with textured inserts on nine different CT scanners with fixed voltage, pitch and computed tomography dose index and extracted 114 texture features with their in-house software. The distribution of the features’ values was different, implying that the variability was related to the CT scanner brand.

Buch et al. [10] also explored an in-house phantom made out of cereal and mayonnaise on two different CT (same brand but different number of detectors 16b and 64b) and demonstrated a significant difference in the computation of the histogram and GLCM features. 

Mahmood et al. [21] used an anthropomorphic lung phantom with shredded rubber and sycamore wood inserts to perform acquisitions on machines from three manufacturers, with a constant voxel size, kVp, pitch and slice thickness. None of the 27 texture features computed with IBEX passed the reproducibility criteria.

#### 3.3.2. Reconstruction Algorithm

Solomon et al. [24] studied three reconstruction algorithms: filtered back projection (FBP), adaptive statistical iterative reconstruction (ASIR) and model-based iterative reconstruction (MBIR). Twenty patients’ diverse conditions were scanned on a single CT scanner, and 23 features were extracted with the three algorithms. Compared with using reference conditions using an FBP reconstruction algorithm at a high dose, between 1 and 3 features for ASIR and between 9 and 11 features for MBIR were affected by a change in the reconstruction algorithm, depending on the organ involved. This suggests a significant impact of the reconstruction algorithm on the texture analysis.

Midya et al. [22] also studied the role of ASIR. They extracted 248 features from CT scans performed on a uniform water phantom, an anthropomorphic phantom and one patient. They observed that an increase in the percentage of ASIR compared to FBP alone increased the blurring of the image and decreased the number of comparable texture features, meaning different ASIR levels could not be mixed in the same study. 

Kim et al. [15] studied the influence on texture features of choosing FBP or sinogram affirmed iterative reconstruction (SAFIRE). They studied lung nodules in 42 patients and extracted 15 features. They showed that among those features, five first order tumour intensity features and four co-occurrence GLCM-based features were significantly affected by the choice of reconstruction algorithm. They however noted that the inter-reader variability induced by the segmentation of the region of interest (ROI) was significantly higher than the one induced by the reconstruction algorithm for nine features but for entropy, homogeneity and the four GLCM-based features, the inter-reconstruction algorithm variability was greater.

### 3.4. Acquisition Parameters

#### 3.4.1. Tube Voltage

Tube voltage sets the number and energy of produced photons and is fixed prior to the acquisition. Fave et al. [13] studied the influence of a change in kVp on 23 features extracted with IBEX from the CBCT scans of 20 patients taken at 120 kVp and 300 mA. Through a simulation algorithm, they explored the effects of 80 kVp, 100 kVp and 140kVp and showed that the intrapatient variability due to a change in voltage was always inferior to the interpatient variability. Buch et al. [10] showed the same results on their in-house phantom when applying 80 kv to 140 kv. Indeed, they studied a set of 42 texture features derived from CT scan images using a custom software. They witnessed no significant statistical variation for any of the included features.

#### 3.4.2. Tube Current

Tube current sets the number of photons and is fixed prior to the acquisition.

Midya et al. [22] performed CT scans with varying tube currents (50 to 500 mA) on a uniform water phantom. The 248 texture features extracted with the in-house software varied with the changes in tube current, particularly when dealing with low tube currents. Mackin et al. [20] extracted 48 texture features from homogeneous and heterogeneous regions of the CT scans of the CCR phantom with a tube current covering 25 to 300 mA. They also concluded that texture features extracted from the homogeneous regions were very dependent on the current value, and had a higher variability than the interpatient variability, while texture features extracted from the heterogeneous regions were less sensible to the current variation. Fave et al. [13] also modified the images obtained on 20 patients (see Section 3.4.1) to simulate different values of tube current, from 100 to 300 mA. They observed that 10 out of 23 texture features had a lower intrapatient variability due to current change than the interpatient variability, but that for 13 out of 23 features, the intrapatient and interpatient variability were of the same order of magnitude. 

On the contrary, Larue et al. [16] scanned the same CCR phantom with nine different CT scanners and different tube currents. The analysis of 114 texture features extracted from a heterogeneous region with in-house software did not reveal a clear influence of the tube current on the texture features but acknowledged that such an investigation deserved to be performed on a larger dataset. Buch et al. [10] also showed no influence of the tube current variation, ranging only from 80 to 120 mAs.

#### 3.4.3. Slice Thickness, Pixel Size

Slice thickness and pixel size both determine the voxel size, which in turn determines the spatial resolution of the image.

Shafiq-ul-Hassan et al. [23], studied the effect of the pixel size and slice thickness on 213 texture features on the CCR phantom. They acquired images with different slice thicknesses and with different fields of view. They subsequently resampled the voxel size to 1 × 1 × 2 mm^3^, and compared the resampled and non-resampled images’ features: 150 were unaffected by the resampling, 42 were significantly improved and 21 were still variable. Larue et al. [16] confirmed their results: they perform CT scans with slice thicknesses of 1.5 and 3 mm on the same phantom and concluded that a large proportion of 114 texture features were affected by the changes and that variability was reduced after resampling the voxel size to 1 × 1 × 3 mm^3^. 

Zhao et al. [26], Lu et al. [17] and Buch et al. [10] conducted the same kind of study; Zhao et al. on a thorax phantom, Lu et al. on 32 patients from the RIDER database and Buch et al. on an in-house phantom, and they all concluded that texture features changed significantly with the slice thickness. Mackin et al. [19] studied the impact of pixel size on intrapatient variability. Their study included eight NSCLC patients and they calculated 150 2.5D texture features (texture features calculated slice by slice then combined) and highlighted that most were dependent on pixel size. They then corrected the differences in pixel size by resampling and filtering, and decreased from 80% to 10% the proportion of features with a higher variability due to pixel size rather than interpatient variability.

#### 3.4.4. Filter

Many filters are provided by CT scanner devices and are named differently according to each brand. Zhao et al. [32] and Lu et al. [17] showed the dependence of texture features on the chosen filter, respectively on a thorax phantom and on 32 patients. The images were reconstructed with both lung and standard filters and they concluded that the chosen filter influenced the value of the texture features.

Mahmood et al. [21] studied 27 texture features extracted from the CT scans of a lung phantom. They only focused on two different kind of secondary order features: neighbourhood grey-tone difference matrix (NGTDM) and GLCM extracted with the IBEX radiomics software. The phantom was scanned on three CT scanners and the images were reconstructed with standard/B40f and lung/B60f filters. They found that none of the features were reproducible when the CT scans were taken with the same manufacturer but reconstructed with different filters.

### 3.5. Contrast Enhancement

Yang et al. [25] studied the dependency of texture features with a time elapse between the injection of the contrast product and the acquisition of the CT scan. They scanned eight patients in two sessions, six times per session, and extracted 122 texture features of lung tumours with IBEX. For seven of their patients, there was no obvious correlation between the time of acquisition and texture features. He et al. [14] extracted 105 texture features using an in-house feature extraction algorithm from 240 CT scans of patients with a lung nodule from both unenhanced and enhanced (25 s after injection) images. They assessed the discriminatory power of each feature using a Mann–Whitney U test in a univariate analysis. They then performed feature selection and dimensionality reduction to build a radiomic signature for each image. Then, they finally analysed the discrimination and classification performance of the radiomic signature and compared the performances for the different sets of CT scans, with or without contrast enhancement. They concluded that UECT gives better results in the discrimination and classification of nodules. 

## 4. Conclusions

This review highlights the variety of effects that changes in acquisition parameters can have on texture features and the difficulty in the interpretation of texture studies. Tube voltage and current appear to have a limited effect on texture features. Tube current was shown to affect heterogeneous regions to a lesser extent than homogeneous ones. Pixel size and slice thickness have a major influence on texture features, highlighting the need for post-processing resampling. The choice of filter also affects texture features and the question of contrast enhanced images is not yet resolved. 

The choice of software, the calculating method (2D or 3D) and the type of CT scanner and brand need to be carefully reported in studies. The question of whether it will be possible through harmonization to get comparable results with CT scanners from different manufacturers is not yet resolved. As studies published outside the scope of this study suggest, the manufacturer variability can be reduced by using a controlled protocol [33]. Moreover, variability also occurs between scans taken on the same CT scanners, and thus asks the question of the accountability of stochastic noise.

This study limited itself to the variability induced by machine related parameters (acquisition and reconstruction parameters). It is important to note that many of the articles reviewed emphasized the importance of human induced variability and in particular the influence of the segmentation of the region of interest, which is seen as a major factor of variability.

This review advocates for the need to state as precisely as possible the methodology regarding the CT scanners (brand, acquisition parameters) used and the post-processing (texture software, if in-house software: definition of algorithms). In CT studies, a harmonization of the acquisition parameters is the key to the future of optimal texture analysis

## Figures and Tables

**Table 1 diagnostics-10-00258-t001:** Characteristics of selected articles. CCR refers to the credence cartridge radiomics phantom, RIDER to the Reference Image Database to Evaluate Therapy Response and NSCLC to non-small-cell lung carcinoma. * Fave et al. do not indicate the number of CT. ^†^ The number is not stated by the authors but the patients come from another study, which included 107 patients.

Reference	Phantom	Patients	Number of CT Devices	Number of Patients	Software	Parameters Studied
Al-Kadi 2009 [7]	No	Lung	2	67	In-house	Repeatability
Balagurunathan 2014 [8]	No	RIDER	2	32	In-house	2D/3D
Berenguer 2018 [9]	Pelvic + CCR copy	No	5	NA	IBEX	Repeatability and redundancy, various acquisition parameters
Buch [10] Caramella 2018 [11]	In-house In-house	No No	1 2	NA NA	LIFEx In-house	Tube voltage, current, slice thickness Repeatability
Fave 2015a [12]	No	NSCLC	? *	20	IBEX	Voltage, current, 2D/3D
Fave 2015b [13]	CCR	NSCLC	19	10	IBEX	Repeatability, CT scanner brand
He 2016 [14]	No	Lung	1	240	In-house	contrast enhancement
Kim 2016 [15]	No	Lung nodule	1	42	In-house	Reconstruction algorithm
Larue 2017 [16]	CCR	NSCLC	9	325	In-house	Repeatability, current, slice thickness
Lu 2016 [17]	No	RIDER	1	32	In-house	Slice thickness, filter
Mackin 2015 [18]	CCR	NSCLC	16	20	IBEX	CT scanner brand
Mackin 2017 [19]	No	NSCLC	1	8	IBEX	Pixel size
Mackin 2018 [20]	CCR	NSCLC	2	107 †	IBEX	Current
Mahmood 2017 [21]	Lung	No	3	NA	IBEX	Filter, CT scanner brand
Midya 2018 [22]	Uniform + anthropomorphic	Abdominal scan	1	1	In-house	Current, reconstruction algorithm
Shafiq-ul-Hassan 2017 [23]	CCR	No	8	NA	In-house	Slice thickness, pixel size
Solomon 2016 [24]	No	Lung, liver, kidney	1	20	In-house	Reconstruction algorithm
Yang 2015 [25]	No	Lung	1	8	IBEX	Contrast enhancement
Zhao 2014 [26]	Thorax	No	1	NA	In-house	Slice thickness, filter

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
