# Peer review of "CT Texture Analysis Challenges: Influence of Acquisition and Reconstruction Parameters: A Comprehensive Review"

_diagnostics, 2020, doi:10.3390/diagnostics10050258_

Round 1
Reviewer 1 Report
This paper was well written and educational.
There was a mark such as * in the table.
However, I cannot found the corresponding meaning.
please add the description.
Author Response
I have added the missing description of the table.
Please see attachement
Thanks
Benoit Charmettant
Reviewer 2 Report
The paper entitled « CT texture analysis challenges…. comprehensive review” is a well-documented and well written review of a the many factors that influence the results of CT texture analysis in the literature. It meets the needs of quite a number of radiologists and clinicians regarding texture analysis. Texture analysis is indeed a “hot topic”. It is more frequently used in research rather than in daily practice because of the variability of its results. Lack of standardization and poor reproducibility are often cited as one of the drawbacks of texture analysis. This is why a review regarding CT texture analysis challenges is of interest. From a more focused point of view:
Abstract and Key words: OK
Introduction: OK
Methods: OK
Results: This section describes and discusses all the potential parameters that may influence the results of texture analysis. It is logically constructed and easy to read. However some minor revisions are to be made:
- All over the results section, there are errors regarding the numbering of the references. For example page 3 line 105, Berenguer et al is cited as [16] while it is [29] in the references section. The same remarks apply pages 4, 5, 6, and 7 (numerous errors)
- There are also errors regarding the names of the authors. For example, page 3 line 72 Buvat et al [7] is cited while Buvat is the last author of paper. Nioche should be cited in place of Buvat.
- Page 3 lines 93 - 95: the sentence is unclear and should be rephrased.
- There are grammatical errors: page 6, line 228: please correct: They scanned 8 patients….
- Page 6 line 245; please correct contrast enhanced…
In conclusion, this paper addresses of the main drawbacks of texture analysis,that is its variability depending on numerous physical parameters. It needs a few minor amendments.
Author Response
I fixed the reference's indexing. I have clarified the sentence p3 and corrected the grammar mistakes.
Please see attachment,
Thank you for the review,
Benoit Charmettant